# Distinct Subcellular Compartments of Dendritic Cells Used for Cross-Presentation

**DOI:** 10.3390/ijms20225606

**Published:** 2019-11-09

**Authors:** Jun Imai, Mayu Otani, Takahiro Sakai

**Affiliations:** Laboratory of Physiological Chemistry, Faculty of Pharmacy, Takasaki University of Health and Welfare, Takasaki, Gunma 370-0033, Japan; nrd15766@gmail.com (M.O.); sakai@takasaki-u.ac.jp (T.S.)

**Keywords:** dendritic cells, cross-presentation, major histocompatibility class I, endoplasmic reticulum-associated degradation, unfolded protein response, inflammation

## Abstract

Dendritic cells (DCs) present exogenous protein-derived peptides on major histocompatibility complex class I molecules to prime naïve CD8^+^ T cells. This DC specific ability, called cross-presentation (CP), is important for the activation of cell-mediated immunity and the induction of self-tolerance. Recent research revealed that endoplasmic reticulum-associated degradation (ERAD), which was first identified as a part of the unfolded protein response—a quality control system in the ER—plays a pivotal role in the processing of exogenous proteins in CP. Moreover, DCs express a variety of immuno-modulatory molecules and cytokines to regulate T cell activation in response to the environment. Although both CP and immuno-modulation are indispensable, contrasting ER conditions are required for their correct activity. Since ERAD substrates are unfolded proteins, their accumulation may result in ER stress, impaired cell homeostasis, and eventually apoptosis. In contrast, activation of the unfolded protein response should be inhibited for DCs to express immuno-modulatory molecules and cytokines. Here, we review recent advances on antigen CP, focusing on intracellular transport routes for exogenous antigens and distinctive subcellular compartments involved in ERAD.

## 1. Introduction

Dendritic cells (DCs) are a diverse group of specialized leukocytes that promote immunity or tolerance by sampling antigens and presenting them to T cells [1]. DCs can also provide immunomodulatory signals through cell–cell interactions and cytokine secretion [2]. In the peripheral tissues, immature DCs constitutively incorporate exogenous proteins. Thereafter, DCs migrate towards the draining lymph nodes, where they process the internalized exogenous proteins and present the antigenic peptides, on major histocompatibility complex class I (MHC I) molecules, to naïve CD8^+^ T cells [2]. The specific ability of DCs to present exogenous antigens—typically presented on MHC class II molecules—on MHC I molecules is called cross-presentation (CP) [2]. CP plays a definitive role in immune response/homeostasis; it can either initiate CD8^+^ T cells to activate an immune response against tumors and/or viruses (cross-priming) or induce peripheral tolerance (cross-tolerance) [3,4,5,6,7,8,9,10]. Despite the importance of CP in shaping the adoptive immune response, the molecular mechanisms underlying this process have remained unclear. Previous studies investigating mechanisms of CP have revealed that, once internalized, exogenous proteins are transported to both the endoplasmic reticulum (ER) and the endosome and processed through the ER-associated degradation (ERAD) pathway, first defined as an arm of the protein quality control system in the ER: The unfolded protein response (UPR). The ERAD pathway specifically recognizes misfolded or unassembled proteins in the ER, which are then retro-transported out of the ER lumen into the cytosol, ubiquitinated, and degraded by the proteasome, to maintain cellular homeostasis [11,12,13]. However, accumulating evidence indicates that both ERAD-dependent processing and CP-mediated peptide loading, are not carried out in the ER, rather in non-classical endocytic compartments, which show distinctive features of the ER [14,15]. For effective ERAD-dependent processing, activation of the UPR is required [11,12,13]. In addition to the classical ERAD substrates (i.e., misfolded and/or unassembled proteins), the accumulation of internalized exogenous proteins in DCs trigger the ERAD pathway and induce the UPR. Since UPR activation in DCs is independent of ER stress in normal conditions [16,17], this suggests that UPR-induced ERAD may play a role in CP [17].

In addition to antigen presentation, DCs detect various environmental signals using pattern recognition receptors (PRRs) and induce T cells to mount an appropriate immune response based on the surrounding conditions, by exchanging immuno-modulatory molecules on the cell-surface and by releasing specific cytokines [18]. A substantial number of PRRs, such as those detecting specific pathogen-derived antigens, immuno-modulatory molecules, and cytokines, are translated via the ER, ER stress should be, in theory, avoided to ensure the maturation of such molecules. However, recent evidence revealed that ER stress alone induced the production of inflammatory cytokines [19,20]. Conversely, the triggering of innate immunity activated the UPR [21] and an excess of ER stress impaired the immuno-regulatory functions of DCs [22]. In contrast, the activation of the UPR was essential for the expression of immuno-modulatory molecules and cytokines by DCs [23].

In this article, we review the current concepts linking CP with the UPR and discuss how the fine tuning of such events allows DCs to perform ER homeostasis and immuno-regulatory functions together.

## 2. DC Subsets

DCs are divided into three main populations: Conventional or classical DCs (cDCs), which are further divided into two subpopulations called T helper 1 (T_H_1) activating cDCs (cDC1s) and T helper 2 (T_H_2) activating cDCs (cDC2s), plasmacytoid DCs (pDCs), and monocyte-derived DCs (moDCs) [24,25].

Since cDC1s mostly activate the T_H_1 response, this DC subset is believed to be dedicated to CP. During the Th-1 response, cDC1s activate CD4^+^ T cells by producing IL-12 [26] and IL-15 [27], through the expression of cell-surface-associated molecules such as CD70, CD40, CD80, and CD86 [28,29,30]. In contrast, cDC1s also attenuate the Th-1 response by B and T lymphocyte attenuator (BTLA) [31,32] and programmed death ligand 1 (PD-L1) [33]. Without stimulation by innate immunity, cDC1s mediate naïve T cell tolerance by BTLA [34] and integrin α_V_β8 with TGF-β [35,36].

cDC2 mainly trigger the T_H_2 response and show restricted CP ability as compared with cDC1s [24,37,38,39]. However, they efficiently present secreted antigenic peptides loaded on MHC I, in the thymus, for central tolerance [40]. Similar to cDC1s, cDC2s produce several different cytokines and express cell-surface-associated molecules to prime naïve CD4^+^ T cells into Tregs, Th2, or Th17 cells [18]. Under steady-state, the UPR pathway can be activated independently of ER stress in cDC1s, but not in cDC2s [16,17], suggesting that the UPR-induced ERAD pathway may play a role in CP [17].

pDCs play an important role in immune defense against viral infections, by secreting large amounts of type I interferon via conventional UPR activation [41]. In view of their restricted role in CP [42,43,44], they will not be discussed in detail in this review.

moDCs, which express CD11c and MHC II [45], differentiate from monocytes and are found to be accumulated in inflamed tissues. For this reason, they are also called inflammatory DCs [46,47,48]. Related to moDCs are the bone-marrow derived DCs (BMDCs), which differentiate from myeloid cells, under the influence of granulocyte-macrophage colony-stimulating factor (GM-CSF). Indeed, both moDCs [49,50,51] and BMDCs [52,53,54,55,56] exhibit efficient CP ability, similar to that of cDC1s, and activate naïve CD8^+^ T cells [57], produce several different cytokines [58], and express cell-surface-associated molecules (e.g., CD70, CD40, CD80, and CD86) [59].

## 3. The UPR

The ER is the site for the synthesis, folding, modification, maturation, and trafficking of both secretory and membrane-associated proteins [60]. The amount of such proteins is approximately one-third all the proteins synthesized in the cell [61,62]. Thus, all the steps involved in the synthesis, processing, and sorting take place in the presence of a high protein concentration, estimated to reach 100 mg/mL, a concentration at which deleterious aggregation is clearly promoted [63]. Moreover, the ER is involved in lipid synthesis and supplies membranes to other cellular compartments [64]. Given these critical roles, ER homeostasis is essential for the viability of eukaryotic cells. The above-mentioned activities are mostly dependent on the protein quality control system of the ER designated as the UPR, which ensures correct folding of the newly synthesized proteins as well as expansion of the ER corresponding to the membrane requirements of the cell [11,12,13].

The stress-induced conventional UPR is composed of three sensors/pathways in mammals, namely activating transcription factor 6 (ATF6), inositol-requiring enzyme 1 (IRE1)-X-box binding protein 1 (XBP1), and protein kinase double-stranded RNA-dependent (PKR)-like ER Kinase (PERK) [20]. Under non-stress conditions, binding immunoglobulin protein (BiP)—also known as glucose-regulated protein (GRP)78—which is an ER-resident molecular chaperone, associates with ATF6, IRE1, and PERK to keep them inactive [20] (Figure 1). When unfolded proteins accumulate, BiP dissociates from these molecular sensors to bind the unfolded proteins [20]. This leads to the activation of all three molecular sensors and the induction of their specific downstream pathways, with the ultimate goal to either restore ER homeostasis or induce cell apoptosis in case of severe, unresolvable ER stress [20] (Figure 1).

ATF6 is a type II transmembrane molecule with a transcription regulatory region in its cytosolic domain [20]. Under ER stress conditions, ATF6 is transported to the Golgi apparatus and processed and the resultant cytosolic fragment acts as a transcription factor [20]. Activated ATF6 induces the transcription of ER-resident molecular chaperones, which accelerate the folding of de novo proteins and increase the capacity of the ER to resolve ER stress [65]. Additionally, ATF6 activates the transcription of XBP1, the transcription factor downstream of the IRE1-XBP1 pathway [66] (Figure 1b).

IRE1 is a Type I ER transmembrane protein with both serine/threonine-protein kinase and endo-ribonuclease activities [67,68]. Under ER stress conditions, IRE1 homo-dimerizes and self-phosphorylates. This conformational change determines the non-conventional splicing of XBP1 mRNA [66,69] (Figure 1c). The resultant XBP1 spliced isoform (XBP1s), acts as a transcription factor and induces the transcription of three groups of molecules to attenuate ER stress: (i) ER-resident molecular chaperones, which accelerate the folding of de novo proteins, (ii) ERAD-related molecules, which are responsible for removing unfolded proteins from the ER [66,70,71,72], (iii) molecules involved in lipid synthesis, which are important for ER membrane expansion [73] (Figure 1c). Additionally, IRE1 shows endonuclease activity against mRNAs and is involved in a mechanism called regulated IRE1-dependent decay (RIDD), which attenuates the translation of ER-resident molecules [74,75] (Figure 1c).

PERK is a Type I ER transmembrane protein that harbors serine/threonine-protein kinase domain in its cytoplasmic region [20]. Under ER stress conditions, PERK undergoes homo-dimerization, autophosphorylation, and activation [20]. The activated PERK phosphorylates the α subunit of eukaryotic translation initiation factor 2 (eIF2α) [76,77]. The phosphorylated eIF2α inhibits the guanine nucleotide exchange factor eIF2B and attenuates translation rate to resolve ER stress [76,77] (Figure 1a). In contrast, phospho-eIF2α promotes the translation of ATF4, a transcription factor, which induces ER chaperones (e.g., GRP78 and GRP94), in order to attenuate ER stress [62]. In conventional UPR, the ATF6, IRE1-XBP1, and PERK pathways orchestrate ER homeostasis by minimizing the consumption of cellular resources. First, PERK phosphorylates eIF2α to prevent translation of newly synthesized proteins into the ER, in order to decrease ER stress [76,77]. Thereafter, ATF6 activates the transcription of ER-resident molecular chaperones, which accelerate the folding of accumulated unfolded proteins to resolve ER stress [65]. Finally, XBP1s initiates translation of ERAD-related molecules, to dispose of the unfolded proteins and the ER-resident molecular chaperones [66,70,71,72]. When such recovery mechanisms fail and ER stress is not resolved, CCAAT/enhancer binding protein homologous transcription factor (CHOP) is translated in an ATF4 dependent manner and the ATF4/CHOP heterodimer induces apoptosis to eliminate the aberrant cells [78,79] (Figure 1d).

Importantly, other than the above-mentioned stress responding functions, the UPR plays essential roles in the development and maturation of immune cells. For instance, both B cells and pDCs, which are highly secretory cells, require activation of the canonical UPR pathway for their development and survival [41,80]. However, in cDC1s, but not in cDC2s, the IRE1-XBP1 pathway is activated independently from ER stress [16,17] and designated as the non-canonical UPR. The non-canonical UPR pathway is essential for the development and survival of cDC1s and BMDCs [41], suggesting that it may play an important role in CP. Nevertheless, little is known about the functions of the non-canonical UPR signaling pathway in immunity and immune disorder pathogenesis.

## 4. Processing of Exogenous Proteins

Previous studies on the molecular mechanisms of CP found two major pathways: The transporter associated with antigen processing (TAP)-dependent pathway and the TAP-independent pathway [81,82]. In the TAP-dependent pathway, internalized exogenous proteins are retro-transported into the cytosol through the cellular membrane, and then processed by the ubiquitin-proteasome system (UPS) into antigenic peptides [81,82]. The second half of this pathway corresponds with the ERAD. Antigenic peptides are transported by the TAP complex into membranous compartments and loaded onto MHC I with the aid of the peptide loading complex (PLC), consisting of TAP1/2, Tapasin, calreticulin, and protein disulfide isomerase (PDIs) [83,84,85]. In the TAP-independent pathway, a lysosomal cysteine protease, cathepsin S, plays a critical role in antigen processing [86]. After processing, antigenic peptides are loaded onto recycled MHC I by a peptide exchange reaction in the endosome/lysosome. In moDCs, both pathways run simultaneously and the selection of the presentation pathway depends on the accessibility for endo/lysosomal proteases, whose activities are dependent on both the maturation of endosomal compartments and the nature of the protein, which, in turn, depends on the uptake route [87]. Exogenous proteins processed by endo/lysosomal proteases undergo TAP-independent pathway, whereas exogenous proteins that escaped degradation in the endo/lysosomes go through the TAP-dependent pathway [87]. Although the TAP-independent pathway has some functions in CP, recent investigation showed that the TAP-dependent pathway plays a major role in this process. Indeed, expression of PLC was found to be higher in splenic cDC1s than in splenic cDC2s [88]. Moreover, results obtained from graft rejection experiments [89], immunoproteasome-deficient BMDCs [90], and enhancement of CP by cytoplasmic delivery of exogenous proteins [91] indicated the significant role of the TAP-dependent pathway in CP. A recent investigation found a proteasome-dependent, TAP-independent pathway in B cells and BMDCs. In this pathway, exogenous proteins were transported into an endosome and the proteasome in the fused autophagosome produced antigenic peptides [92,93]. In addition to the proteasome-dependent, TAP-independent pathway, several other CP-related pathways were found in DCs, such as the membrane rupture pathway [94]. However, as the initiation of naïve CD8^+^ T cells by such pathways was not verified under physiological conditions, they will not be discussed further in this review.

## 5. Internalization and Transport of Exogenous Proteins

The molecular mechanisms of exogenous protein transport, upstream of processing by the UPS in CP, are not known, in part due to contrasting experimental results. This relies at least in part on the source of exogenous proteins (e.g., soluble proteins, bead-bound proteins, or proteins expressed by heat-killed microbes), which might vary among different experiments, as well as on their uptake route (e.g., receptor-mediated endocytosis, pinocytosis, and phagocytosis). In addition, the particle size of the antigen-loaded beads also influences the intra-cellular transport pathways of an identical antigen [95]. Moreover, although experiments aimed at investigating CP were carried out in cDC1s, moDCs or BMDCs, all of which show high CP efficiencies, the intracellular transport mechanisms might not be the same. cDC1s showed the most substantial CP efficiency, irrespective of the protein uptake route [37]. The CP efficiency of cDC2s was equivalent to cDC1s for receptor-mediated endocytosis but less efficient for pinocytosis and phagocytosis [37]. moDCs showed equivalent CP efficiency to cDC1s for pinocytosis, less efficiency for receptor-mediated endocytosis, and no detectable CP for phagocytosis [37]. For instance, Rab43 was required for retrograde transport from the endocytic pathway to the Golgi apparatus and highly expressed in cDC1s, but less in cDC2s and moDCs [96,97]. Moreover, Rab43 was essential for CP of cell-associated, but not soluble proteins, by cDC1s. In contrast, Rab43 was dispensable for CP of both cell-associated and soluble proteins by cDC1s and cDC2s, as well as by moDCs [96]. WD repeat- and FYVE domain-containing protein 4 (WDFY4), which plays an important role in proper subcellular vesicular targeting, was specifically required for CP of cell-associated proteins by cDC1s [38]. Insulin-responsive aminopeptidase (IRAP), which colocalized with the endosomal markers Rab14 and syntaxin 6—both known to be associated with regulated endosomal storage compartments—played an important role in CP of both soluble and insoluble proteins by cDC1s [39], and was essential for CP of soluble proteins by moDCs [25], but not by cDC2s [39]. In BMDCs, TLR4 and Fc receptor ligation resulted in the interaction of Rab14 with the kinesin KIF16b, accelerated anterograde transport of endosomes and phagosomes, and delayed fusion with lysosomes, resulting in the promotion of CP [98].

Altogether, such observations clearly indicate that the molecular machinery underlying intracellular protein transport differs among distinct DC subsets. Recent studies have demonstrated that exogenous proteins are transported into non-classical endocytic compartments, processed by the ERAD-dependent machinery, retro-transported out of lumenal fractions, and processed by the UPS; although, the exact mechanisms of internalization have not been elucidated [14,15,99]. DCs protect exogenous proteins from degradation by lysosomal proteases, and this ability is inversely proportional to the CP ability of each DC subset. cDC1s express lower amounts of lysosomal proteases compared with cDC2s [100] and moDCs [100]. Moreover, lower expression of lysosomal proteases with protease inhibitors was observed in DCs as compared with macrophages (M*φ*) both in vivo (spleen and lymph node) and in vitro (bone marrow-derived dendritic cells (BMDCs) and bone marrow-derived macrophages (BMM*φ*) [101]. The level of cathepsin S, L, K, B, D, E, H, and O and asparagine endopeptidase was lower in BMDCs as compared with BMM*φ*, which resulted in the preventing degradation of the exogenous proteins [101]. Inhibition of cathepsin translation due to loss of YTH N6-methyladenosine RNA binding protein 1 (YTHDF1), enhanced CP in cDCs [102]. The maturation kinetics of phagosome was faster in BMM*φ* than in BMDCs [103]. In BMDCs, phagosome/endosome maturation was delayed after TLR4 ligation, which in turn promoted CP by downregulating vacuolar proton ATPase, cathepsin B, D, S, and Rab7 [98,103,104] and upregulating MHC I, PLC, and UPS [104]. In cDCs, activation of NOD1 and NOD2 also accelerated CP by upregulating PLC and ERAD-related molecules [105]. In human moDCs, NOD and TLR2 activation enhanced CP by positively regulating MHC I peptide loading and immunoproteasome stability [106].

To protect antigenic peptides from degradation by lysosomal proteases, DCs utilize other methods. For instance, BMDCs maintain phagosomes and endosomes/lysosomes under an alkaline pH (7.5–8) [107], while Mφ) and neutrophils maintained them at pH 4.5–7 [107,108], to inhibit activation of lysosomal proteases. The high pH of the phagosome and endosome/lysosome was attributed to reduced V-ATPase activity [109] and recruitment of nicotinamide adenine dinucleotide phosphate (NADPH) oxidase NOX2 [107,110] at very high rates (mM/s) [111]. Reduced V-ATPase activity impairs proton-transport into the luminal space, resulting in moderate acidification. Increased NOX2 produces reactive oxygen species (ROS), which react with the protons in the luminal space, thereby creating an alkaline environment [111]. NOX2 is made up of six subunits, Rac1 or Rac2, gp91phox (containing heme), p22phox, p40phox, p47phox, and p67phox [111]. Active alkalization by NOX2 was seen to be regulated by Rab27a [110], a plasma membrane SNARE protein called VAMP-8 (in both BMDCs and human moDCs) [112], phagosomal SNAREs syntaxin-4, and SNAP-23 (in BMDCs) [113]. Rac2 regulated the recruitment and the assembly of NOX2 in cDC1s but not cDC2s [107]. The deletion of Wiskott–Aldrich syndrome protein (WASp) increased Rac2 activity, which resulted in enhanced CP efficiency, both in cDC1s and cDC2s [114]. In contrast, the reduced activities of either gp91phox or p47phox impaired the CP ability of BMDCs [107]. In cDC1s, sialic acid-binding immunoglobulin-type lectin-G (Siglec-G), a member of the lectin family, recruited Src homology region 2 domain-containing phosphatase-1 (SHP-1) to dephosphorylate p47phox, which inhibited NOX2 activation in the phagosomes [115]. In BMDCs and cDC1s, transcription factor EB (TFEB) negatively regulated CP by up-regulating lysosomal proteases and promoting the maturation of lysosomes [116]. Moreover, pharmacological inhibitors of endocytic acidification, (i.e., chloroquine and ammonium chloride) and lysosomal protease inhibitors (i.e., leupeptin), accelerated CP in human moDCs and cDC1s [87,117,118] and murine BMDCs [119]. All these results indicate that escape from endo/lysosomal proteases is important for effective CP.

The retro-transport of internalized protein from endocytic compartments to the cytosol is another important process required for efficient of CP. The ability to retro-transport was proportional to CP efficiencies among different DC subsets [100]. In vivo and in vitro studies showed that ERAD plays an important role in retro-translocation. Indeed, ER-resident molecules, including the ERAD machinery, were found in the phagosome of both Mφ) [120,121] and cDCs [122] and in the non-classical endocytic compartments of cDCs, moDCs, and BMDCs [14,15]. Expression of ERAD-related molecules (i.e, calreticulin, calnexin, SEC61α, SEC61β, SEC61γ, and PDIs) was higher in cDC1s as compared to cDC2s [88]. In contrast, inhibition of valosin-containing protein (VCP) and SEC61 recruitment, via either PYR-41 or thalidomide-mediated NF-κB inactivation, restrained CP in BMDCs [123].

However, the molecular mechanisms of ERAD-dependent processing in CP are not known.

## 6. ERAD-Dependent Processing in CP

In the ERAD pathway, misfolded proteins are recognized in the ER lumen, retro-translocated to the cytosol, and rapidly degraded by the UPS [20]. These steps are tightly regulated because regardless of whether the protein is misfolded or unassembled, secretory proteins cannot stay folded under cytosolic conditions and therefore easily aggregate with properly folded cytosolic proteins, thereby becoming highly toxic for the cells. In CP, as some exogenous proteins are derived from infectious pathogens, apoptotic cells, or cancer cells, stringent ERAD management is required.

### 6.1. Substrate Recognition

Based on the recognition site of unfolded proteins, the ERAD pathways are further subdivided into three groups: ERAD-M, ERAD-L, and ERAD-C, which recognize misfolding in the transmembrane, luminal, and cytosolic region, respectively [124]. Moreover, depending on the localization of the recognition region, the retro-transport machinery is different for these three pathways. However, since almost all proteins involved in CP are ERAD-L substrates, we focused on the ERAD-L pathway.

In the ERAD pathway, recognition of unfolded or misfolded proteins has to be strictly controlled, because degradation of properly folded proteins might be wasteful [125]. Misfolding, glycosylation, and incorrectly formed disulfide bonds are common features for substrate recognition. Misfolded proteins are recognized by ER-resident molecular chaperones, such as BiP and GRP94 [20]. Oligosaccharides are distinguished by ER-resident mannose-specific lectins, such as ER-degradation enhancing α-mannosidase-like proteins (EDEMs) and amplified in osteosarcoma 9 (OS-9) [126,127,128], whereas incorrectly formed disulfide bonds are identified by redox-driven PDIs [129,130,131]. Some of the characteristics for the ERAD substrate are also observed in the substrates for CP [132], because the processing of correctly folded intracellular proteins results in direct presentation, which competes with CP. Although the precise molecular mechanism for the identification of exogenous proteins in CP is not known, the fact that UPR—which up-regulates ER-resident chaperones—is necessary for efficient CP, would suggest a critical role for ER-resident chaperones in CP [55].

#### 6.1.1. BiP

The ER-resident heat shock protein (Hsp)70 family member BiP, can interact with both glycosylated and non-glycosylated proteins [133,134]. BiP preferentially binds the hydrophobic region of newly translated proteins and chaperones their folding. The unfolded proteins expose their hydrophobic core causing BiP to strongly associate with them. If the affinity of the substrate is too strong to dissociate from BiP after the refolding process, the substrates are perceived to be misfolded and retro-transported to the cytosol, with the aid of endoplasmic reticulum-localized DnaJ homologous 5 (ERdj5) and suppressor of lin-12-like 1 (SEL1L) [135]. Thus, BiP plays a central role in ERAD for both misfolded and unassembled proteins (Figure 2).

In CP, BiP specifically interacted with internalized exogenous proteins [132], indicating that BiP also played an important role in the recognition of exogenous proteins and that these proteins were unfolded before arriving at the luminal membrane. The enhanced CP efficiency of BiP-bound proteins suggests that recognition by BiP is a critical step in CP [136]. Moreover, Hsp-complexed proteins show high CP efficiency, supporting the importance of the unfolding of exogenous proteins for recognition by BiP in CP [137] (Figure 2).

#### 6.1.2. Mannose-Specific Lectins

The ERAD system uses the sugar chains of glycoproteins to monitor the conformational maturation of the protein and decides whether to direct them to the ERAD as misfolded proteins [126,127,128,138,139]. ER-resident glycoproteins are co-translationally modified by high-mannose glycans, with the structure Glc_3_Man_9_Glc-NAc_2_ (glucose (Glc), mannose (Man), and N-acetylglucosamine (GlcNAc)), on conserved asparagine residues of the N-glycosylation motif (NxS/T) [140,141]. Terminal sugars of high-mannose core glycans is removed by ER-resident exo-glycosidase (i.e., glycosidase I and II and ER mannosidase I) in a time-dependent fashion. While the high-mannose core retains 9 mannose residues (Man_9_Glc-NAc_2_), glycoproteins undergo a folding process with the aid of lectin-type chaperones, such as calnexin or calreticulin. After the removal of additional mannose residues in the ER, the high-mannose core (Man_5~7_Glc-NAc_2_) of glycoproteins are recognized by EDEMs as the ERAD substrate [126,127,128,138,139]. This occurs because the processed high-mannose cores (Man_5~7_Glc-NAc_2_) do not fold properly in time in the ER and, consequently, go through the ERAD process with the aid of other lectins, such as OS-9 and XTP3-B [142,143] (Figure 2). In contrast to misfolded glycoproteins, folded glycoproteins are exported to the Golgi apparatus, where the high-mannose cores of folded glycoproteins are removed by Golgi-resident mannosidase I, to produce the Man_5_Glc-NAc_2_ core for further glycosylation [144]. However, not all Man_5_Glc-NAc_2_ cores are further processed into complex oligosaccharide chains and remain as high mannose type oligosaccharide chains of Man_5_Glc-NAc_2_ [144].

In CP, considerable amounts of exogenous proteins are mature glycoproteins with high mannose type oligosaccharide chains of Man_5_Glc-NAc_2_ and are preferentially recognized as substrates for the ERAD by EDEMs. Thus, it is possible that high mannose type oligosaccharide chains function as signals of exogenous proteins. In support of this view, our findings report the presence of ERAD-lectins in purified microsomal fraction for CP (personal unpublished data). In the same context, the association of these lectins with exogenous proteins (personal unpublished data) also suggests that these molecules may exert essential roles in CP.

Exogenous proteins that bind the mannose receptor (MR, CD206 or MRC1), are efficiently retro-translocated from the ER to the cytosol [145]. Since the MR recognizes sulfated and mannosylated sugars [146], it is possible that MR substrates are also recognized as glycosylated ERAD substrates after internalization, thereby increasing CP efficiency. The fact that MR-deficient BMDCs showed poor CP ability for glycosylated proteins, supports this hypothesis [145]. The role of sugar chains in CP are now under investigation, although evidence suggests that optimum glycosylation might function as a useful adjuvant for CP [147] (Figure 2).

#### 6.1.3. PDIs

PDIs, characterized by the presence of a thioredoxin-like domain (CXXC active-site motif) and chaperone activity, are the primary protein oxidases in the ER lumen [129,130,131]. The two cysteines of the thioredoxin-like domain are redox-active: an oxidized (disulfide-bonded) or reduced (free) form. PDI family proteins chaperone the formation of correct disulfide bonds, by introducing, reducing, or isomerizing disulfide bonds on substrate proteins, with the aid of ER-resident molecular chaperones [131,148]. In ERAD, PDIs recognize incorrect disulfide bonds on unfolded proteins [129,130]. Indeed, proteins with incorrect disulfide bonds expose their hydrophobic cores, which preferentially bind with ER-resident molecular chaperones, including PDIs. These proteins, which hardly dissociate from PDIs, are recognized as ERAD substrates (Figure 2).

In CP, highly oxidizing conditions in the endosomes foster incorrect disulfide bond formation in the exogenous protein, which is preferentially recognized by PDIs. In this context, it might be possible that gamma-interferon-inducible lysosomal thiol reductase (GILT), the only known thiol reductase localized in the lysosomes and phagosomes, is essential for CP in BMDCs [149], suggesting a critical role for disulfide bond formation for substrate recognition in ERAD-dependent processing.

It is worth noting that PDIs associated with incorporated exogenous proteins [132] and also localized in our purified microsome for CP [15], further suggesting that incorrect disulfide bonds might be a marker for the identification of exogenous proteins (Figure 2).

Since ERAD and ERAD-dependent processing in CP share several cellular machineries, understanding the mechanism of ERAD-substrate recognition might be useful for improving CP efficiency.

### 6.2. Retro-Transport of Substrates

Together with retro-translocation, the ERAD substrates are ubiquitinated and degraded by the UPS. The retro-translocation of ERAD substrates is largely reduced by inhibition of ubiquitination, thus indicating a strong link between the two processes [133].

#### 6.2.1. HRD1 Complex in ERAD

HRD1 complex is the most well characterized retro-translocon machinery for ERAD-L. Substrates of the HRD1 complex are first recognized by either BiP (non-glycosylated) or OS-9 (glycosylated), transferred to a core complex consisting of SEL1L homocysteine-induced endoplasmic reticulum protein (HERP), degradation in endoplasmic reticulum protein (DERLIN)-1 (DERLIN-2, DERLIN-3), OS-9 and XTP3-transactivated gene B protein (XTP-3B), ubiquitin-conjugating enzyme E2 G2 (UBE2G2), and ancient ubiquitous protein 1 (Aup1) [134,150,151,152,153,154,155,156,157], and then ubiquitinated by HRD1 [158]. Moreover, HRD1 itself contributes to form the ERAD channel [159]. After poly-ubiquitination, VCP binds the poly-ubiquitin chain with the aid of cofactors (i.e., nuclear protein localization protein 4 (NPL4) and ubiquitin recognition factor in ER-associated degradation protein 1 (UFD1) and then extracts the substrates from the retro-translocon in an ATP-dependent manner [160,161] (Figure 2).

#### 6.2.2. SEC61 Complex in ERAD

The SEC61 complex (SEC61α, SEC61β, and SEC61γ) is a core component of the translocon spanning the ER membrane [162,163]. Newly synthesized secretory and membranes proteins are translocated through this complex [162,163]. Moreover, the SEC61 complex functions as a retro-translocon in ERAD [13]. The 19S regulatory subunit of the proteasome shows ATP-dependent unfoldase activity [164] and extracts ERAD substrates from the ER membrane through the Sec61 complex [124,165]. VCP can also bind to the SEC61 complex and extract ERAD substrates [166,167] (Figure 2).

#### 6.2.3. Translocon in CP

In CP, SEC61α and β are specifically associated with exogenous proteins in moDCs and BMDCs [132]. Inhibition of the SEC61 complex impaired CP ability in BMDCs [121], moDCs [122,132], cDC1s, and cDC2s [14]. The inhibition of SEC61 recruitment into the endosome strongly inhibited CP by cDC1s, but not by cDC2s, indicating that the ERAD-dependent processing of exogenous proteins was carried out not in the ER or the classical endosome, but in an endosome with ER-resident molecules for effective CP of cDC1s [14] (Figure 2). However, the inhibition of the SEC61 complex following sustained treatment with mycolactone, a specific inhibitor of the SEC61 complex, decreased CP efficiency in the cDC1-like mouse cell line MutuDC, indicating that the SEC61 complex was not essential for retro-translocation in CP [168]. The contradictory results strongly suggest the need for further experimentation for clarifying the role of the SEC61 complex in CP. Moreover, the inhibition of endoplasmic reticulum protein 1 (DERL1) degradation showed no effect in CP by both BMDCs and moDCs [14]. Similarly, inhibition of HRD1 resulted in mild impairment in CP by BMDCs [99].

This may be because the non-canonical UPR pathway plays an important roles in the differentiation and function of cDC1s and moDCs [41]. The non-canonical UPR pathway relies on the activation of the IRE1-XBP1 pathway, in which induction of the SEC61 complex is higher than in the HRD1 complex [70]. However, the IRE1-XBP1 pathway up-regulates ER-related secretory pathway genes other than the ERAD-related genes. The SEC61 complex differs from the HRD1 complex in that it works in both directions—as a translocon for protein import and as a retro-translocon for protein export (Figure 2). The SEC61 complex plays a major role in CP because it may be more available under ER stress conditions owing to its ability to channelize protein translocation bidirectionally. Moreover, it might be possible that since the majority of the substrates of the ERAD-dependent processing in CP are simple unstructured proteins, which are unfolded after internalization into DCs and have no trans-membrane region, they might be preferentially exported through the SEC61 complex [124].

In this context, it is worthwhile to note that components of the HRD1 complex were found by our group in purified microsome for CP (personal unpublished data), suggesting that the retro-translocon in CP does not rely on a specific complex, such as the SEC61 complex. Taken together, these observations strongly suggest that DCs may utilize different retro-translocon machineries for CP depending on the condition of exogenous proteins. However, further investigation is required to confirm this assumption.

### 6.3. ERAD-Related Molecules

In addition to the ERAD machinery, a small number of molecules have been shown to play a role in CP. VCP was specifically associated with exogenous proteins in moDCs and BMDCs [132], and the inhibition of VCP also abrogated the CP ability in such DC subsets [132,145,169]. Similarly, carboxyl terminus Hsp70/90 interacting protein (CHIP) associated with CP substrates and played an essential role in CP in moDCs [132]. Tumor susceptibility gene 101 (TSG101), a dominant-negative regulator of poly-ubiquitination [170], co-localized with CP substrates and negatively regulated CP efficiency in BMDCs [145]. Cytosolic molecular chaperones, such as Hsp70 and Hsp90, were associated with CP substrates and required for effective CP [171,172,173]. Altogether, these results indicate shared molecular mechanisms in both ERAD and CP (Figure 2).

Moreover, for effective CP by ERAD-dependent processing, DCs are equipped with an endosome with ER-resident molecules, designated as the non-classical endosome [14,15]. This compartment might be critical for DCs to mediate effective CP and immuno-regulatory functions by avoiding unwanted activation of the UPR. It might also be possible that the non-canonical UPR pathway in DCs plays essential roles to prepare the ERAD-related molecules for this compartment. However, despite these advances, further investigation is necessary to clarify the molecular mechanism of CP and the role of the non-canonical UPR, and the non-classical endosome pathway in CP.

## 7. The UPR Induces Inflammation

DCs detect various environmental signals by their PRRs and induce T cells to mount an appropriate immune response via both immuno-modulatory molecules and cytokines [18]. In addition to the canonical stress response functions, the UPR induces inflammatory cytokine production, independently from the pathogen-triggered innate immunity activation, in many kinds of cells, including DCs (Figure 3).

Moreover, inflammatory cytokines, such as IL-1β and IL-6, were able to induce ER stress and activate the UPR [174] (Figure 3). For instance, TNF-α activated both the IRE1-XBP1 and the PERK pathways [175]. Ligation of PRR activated the UPR in Mφ) and induced production of inflammatory cytokines, such as TNF-α, IFN-β, and IL-6 [176]. In BMDCs, ER stress was found to induce IL-6 [177]. Additionally, in DCs, ER stress and PRR ligation enhanced IL-23 expression [178] and the production of TNF-α, IL-6, and IFN-β [179]. These cytokines play essential roles in host defense, however, their aberrant induction might result in a pathologic inflammation [180]. Although the precise molecular mechanism/s underlying the contribution of pathogenic DCs in chronic inflammatory diseases, such as Crohn’s disease and type 2 diabetes, are not known, mutations in *IRE1* and *XBP1* are supposed to be among the risk factors for such conditions [181,182]. In cDCs, dysregulated activation of Xbp1s induced aberrant triglyceride synthesis and resulted in impaired immunoregulatory functions [22] (Figure 3). More specifically, in cDC1s, over-activation of the IRE1-XBP1 pathway, resulted in a decreased expression of tapasin by the RIDD and hampered CP [16], which might inhibit the direct presentation of intracellular proteins under steady-state conditions (Figure 3). In murine herpes simplex virus (HSV) type 2 infection experiments, deletion of viral glycoprotein D resulted in decreased ER stress and suppressed BMDC functions, such as migration and initiation of naïve T cells, indicating that the over activation of UPR impairs both antigen presentation and immunoregulatory activity of DCs [183]. Although the signal transduction pathway is not known, accumulating evidence shows that tumor UPR can induce UPR in tumor-infiltrating BMDCs in a cell-extrinsic manner (Figure 3). This ER stress transmission to receiver BMDCs up-regulated the production of inflammatory cytokines, such as IL-6, IL-23, and TNF-α and the immunosuppressive enzyme arginase, leading to a proinflammatory/suppressive phenotype [184]. In contrast, activation of the non-canonical UPR in cDCs was essential for the expression of CD80, CD86, MHC II, and inflammatory cytokines [23] (Figure 3). Activation of NF-κB by the UPR was found in several kinds of cells and resulted in the production of inflammatory cytokines. However, the activation of NF-κB was also required for efficient CP through the recruitment of VCP and SEC61 in BMDCs [123]. These results indicate that unregulated activation of the UPR hampers the function of DCs and appropriate activation of the UPR is essential for the CP activity of DCs. Such observations suggest that DCs are likely equipped with the molecular machinery to control the UPR in response to their needs and the external environment.

Recent reports revealed some molecular mechanisms linking UPR and inflammation. Although the related results are not fully obtained in DCs, the molecules involved in such mechanisms are conserved in DCs. Activation of ATF4, ATF6, or XBP1 directly induced transcription of inflammatory cytokines [176,185] (Figure 3). Activated IRE1 induced the expression of inflammatory cytokines by associating with TRAF2 to activate JNK [186] and NF-κB [187], or by associating with stimulator of interferon genes [188]. Additionally, the over-activation of IRE1 attenuated the amount of IκB by RIDD, thus resulting in NF-κB activation and production of inflammatory cytokines [189] (Figure 3). RIDD also produced cytosolic RNA fragments, which directly activated RIG-1 that, in turn, triggered the production of inflammatory cytokines [190] and type-I interferon [191]. These results suggest that intersections of the UPR with inflammatory pathways occur in different cell populations, playing important roles in immunity.

Little is known about the inhibitory system that suppresses the over-activated UPR in DCs. GADD34 is one of the negative feedback factors for the PERK pathway, translated by ATF4 together with CHOP, which dephosphorylates eIF2a as a phosphatase 1 cofactor to attenuate the PERK pathway. Interestingly, GADD34 was expressed in steady-state DCs [192] and conferred strong resistance against severe ER stress in PRR stimulated DCs [193,194,195]. In 293T cells, GADD34 inhibited IKK and NF-κB activation, thus causing suppression of inflammatory cytokine transcription [196]. Thus, GADD34 may represent one of the regulatory mechanisms used by DCs to restrict unwanted UPR activation (Figure 3). Although current knowledge does not provide sufficient clues to clarify the molecular mechanisms that regulate UPR activity in DCs, the non-canonical UPR might constitute one element of such a regulation. However, several questions remain to be addressed in future research.

## 8. Conclusions

Antigen presentation, especially CP, and immunoregulatory functions are two indispensable roles of DCs in the adaptive immune system. Although the UPR plays essential roles in these two pathways, the directions are inverse. For immunoregulatory functions, the type and amount of secreted cytokines and the expression of immunomodulatory molecules, are essential. Therefore, the ER condition is optimized for both protein folding and maturation. In contrast, for effective CP, unfolded exogenous proteins accumulate in the lumenal fraction and are rapidly degraded in an ERAD-dependent fashion. Accumulation of unfolded proteins in the ER-related fractions induced the UPR and hampered the translation and folding of all ER transit proteins. Additionally, recent research showed that activation of the UPR induced inflammation and influenced the direction of adaptive immunity. Indeed, the unregulated activation of the UPR in DCs suppressed anti-tumor immunity [22,184] and over-activation of the UPR was associated with autoimmune and inflammatory disorders such as diabetes, atherosclerosis, and myositis [181,197,198]. However, the role of UPR-mediated DC modulation for each pathogenesis is not known. What we do know is that, to avoid these difficulties, DCs can carry out ERAD-dependent processing, not in the ER, rather in non-classical endocytic compartments with ER-resident molecules [14,15]. Consequently, the non-canonical UPR might supply ERAD-related molecules into such compartments thus enabling DCs to cope with contradictory requirements for ER homeostasis.

In conclusion, although further experimentation is needed to comprehensively address the biology of non-classical endocytic compartments and non-canonical UPR, evidence to date strongly suggests they might play a relevant role in the orchestration of antigen presentation, mainly CP, and immunoregulatory function of DCs.

## Figures and Tables

**Figure 1 ijms-20-05606-f001:**
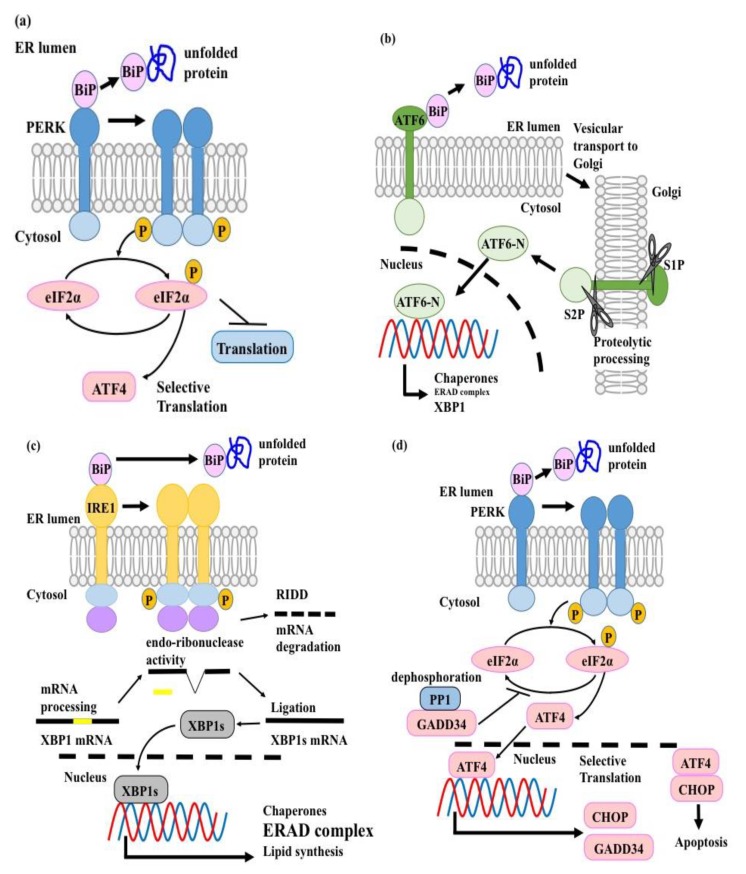
Signal transduction of the unfolded protein response (UPR). Under normal conditions, all three endoplasmic reticulum (ER) stress sensors (i.e., protein kinase double-stranded RNA-dependent (PKR)-like ER Kinase (PERK), activating transcription factor 6 (ATF6), and inositol-requiring enzyme 1 (IRE1)) are associated with binding immunoglobulin protein (BiP). When unfolded proteins accumulate in the ER, BiP dissociates from PERK, ATF6, and IRE1 to bind with the unfolded proteins. As a result, the three molecular sensors are activated: (**a**) PERK forms a homo-dimer complex and phosphorylates eIF2α, which attenuates translation. This inhibits the incorporation of newly synthesized proteins into the ER. In parallel, the selective translation of the transcription factor ATF4 is carried out. (**b**) ATF6, which has dissociated from BiP, is transported into the Golgi apparatus and processed by the Golgi-resident endoproteases S1P and S2P. The resultant ATF6 cytosolic fragment acts as a transcription factor and activates the transcription of ER-resident molecular chaperones, which accelerate the folding of accumulated unfolded proteins. (**c**) Activated IRE1 homo-dimerizes and self-phosphorylates. This conformational change determines the nonconventional splicing of X-box binding protein 1 (XBP1) mRNA. The resultant XBP1 spliced isoform (XBP1s) acts as a transcription factor and induces the transcription of three groups of molecules. (i) ER-resident molecular chaperones. (ii) ERAD-related molecules, to remove unfolded proteins from ER. (iii) Lipid synthesis related molecules. IRE1 also shows regulated IRE1-dependent decay (RIDD) activity to decrease the translation of ER-resident proteins, after completing nonconventional splicing of the XBP1 mRNA. (**d**) If ER stress is too severe to be suppressed by the above-mentioned pathways, ATF4 acts as a transcription factor and induces the transcription of CCAAT/enhancer binding protein homologous transcription factor (CHOP) and GADD33. The ATF4/CHOP heterodimer induces apoptosis to eliminate stressed cells. GADD33 negatively regulates the PERK pathway, by dephosphorylating eI2Fα, to delay the activation of ATF4/CHOP. The dashed line indicates the nuclear membrane.

**Figure 2 ijms-20-05606-f002:**
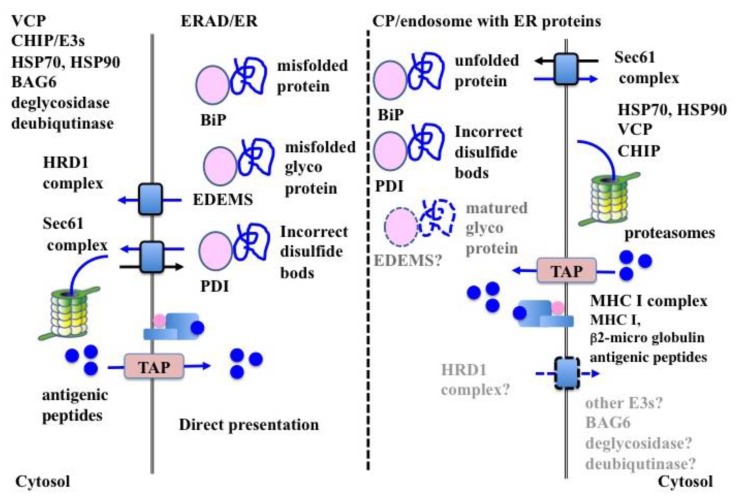
The molecular mechanisms of endoplasmic reticulum-associated degradation (ERAD) and cross-presentation (CP). The processing of exogenous proteins in CP involves a part of the molecular machinery of ERAD; however, significant differences exist. ERAD is carried out in the ER, while ERAD-dependent antigen processing during CP takes place in the non-classical endosomes, together with ER-resident molecules. The recognition of unfolded proteins by ER-resident molecules is similar in both processes, except for the EDEMs. Both the HRD1 and the Sec61 complexes are utilized in ERAD, but the HRD1 complex is not essential for CP. Additionally, several kinds of ERAD-related molecules, such as E3s, cofactors of VCP (i.e., NPL4 and UFD1), co-chaperones of Hsp70 (i.e., Bag6), deglycosidase, and deubiquitinase, play an important role in ERAD but are not investigated in CP. Solid arrows indicate mechanisms supported by experimental evidence. Dashed arrows indicate hypothetical mechanisms that still lack experimental evidence.

**Figure 3 ijms-20-05606-f003:**
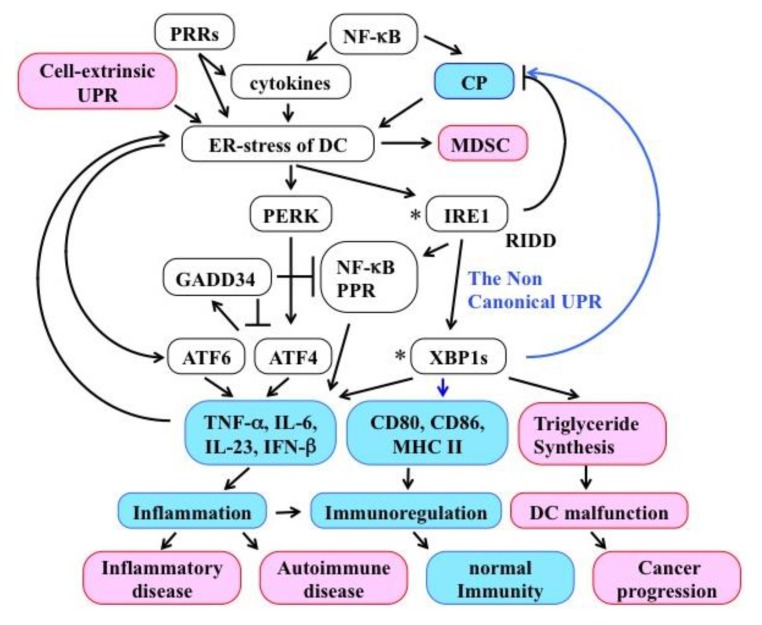
The crosstalk between CP, UPR, and inflammation. Pathogen-related stimuli, such as pattern recognition receptor (PRR) ligation, cytokines, and cell-extrinsic UPR can induce ER stress in Dendritic cells (DCs). Accumulation of unfolded proteins for CP also contributes to trigger/sustain ER stress in DCs. Although ER stress activates the UPR to restore ER homeostasis, the UPR itself accelerates the transcription of several inflammatory cytokines both directly, via UPR-related transcription factors, and indirectly, through the activation of different signaling cascades (e.g., NF-κB, PRRs). When the UPR of DCs is over-activated, the immunoregulatory functions of DCs are dismissed and this may result in chronic inflammation and ultimately cancer. However, when the UPR is over-repressed, CP efficiency is impaired. Thus, the non-canonical UPR of DCs has to cope with opposite requirements in order to maintain both ER homeostasis and adequate CP/immune-modulatory functions. The precise molecular mechanisms to evade UPR over-activation are, however, not known, except for the established suppressive function of GADD33. Black arrows indicate activation, T-arrows indicate suppression, blue arrows indicate pathways involved in the non-canonical UPR. The asterisks of IRE1 and XBPs indicate the risk factor for inflammatory disease among the UPR-related molecules.

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
