# Peer review of "Distinct Subcellular Compartments of Dendritic Cells Used for Cross-Presentation"

_ijms, 2019, doi:10.3390/ijms20225606_

Round 1

Reviewer 1 Report

The overall information included in this review have improved substantially, including more recent studies, and the new figures are more informative. However, I stand by my previous comments regarding the confusing English language and incorrect grammar. A sentence should have a subject, a verb and express a complete thought. While this manuscript is rigged with unfinished sentences, repeated sentences or sentences missing crucial parts. There is also incorrect use of plurals in every other sentence.

The subject matter of the mechanisms underlying the dendritic cell cross-presentation is of high interest and very topical. It is also very complex and the difficult to understand communication of the subject occludes the understanding of this review and decreases its impact severely.

My advice would be to seek professional English editing with careful attention to convey the accurate meaning of the scientific concepts.

Unfortunately, I cannot recommend this manuscript for publication at this stage.

Author Response

Response to Reviewers’ Comments

Reviewer 1:

The overall information included in this review have improved substantially, including more recent studies, and the new figures are more informative. However, I stand by my previous comments regarding the confusing English language and incorrect grammar. A sentence should have a subject, a verb and express a complete thought. While this manuscript is rigged with unfinished sentences, repeated sentences or sentences missing crucial parts. There is also incorrect use of plurals in every other sentence. The subject matter of the mechanisms underlying the dendritic cell cross-presentation is of high interest and very topical. It is also very complex and the difficult to understand communication of the subject occludes the understanding of this review and decreases its impact severely.

My advice would be to seek professional English editing with careful attention to convey the accurate meaning of the scientific concepts. Unfortunately, I cannot recommend this manuscript for publication at this stage.

Reply: We wish to thank the reviewer for his/her positive comments. To address the reviewer’s concerns about the confusing English language and incorrect grammar, we have sought a professional English editing service. The manuscript has been completely rewritten by an expert who paid careful attention to convey the accurate meaning of the scientific concepts.

Reviewer 2 Report

In this manuscript, Imai et al review recent advances on antigen cross-presentation, focusing on the intracellular transport routes for exogenous antigens and the distinctive subcellular compartments involved in the endosplasmic reticulum associated degradation. The paper is clearly written, interesting and timely.

I have a few specific points to address:

1/ Page 5, para 6: Here, the authors should also discuss the fact that antigen export is also influenced by the size of the phagocytosed antigen ((see for example this publication: Mant et al. Immunology. 2012 jun; 136(2):163-75).

2/ Page 6, para 6: In this section, the authors should be more moderated concerning the contribution of Sec61 in antigen cross-presentation. The precise involvement of Sec61 remains unclear and fraught with technical issues (for discussion, see this publication: Romisch K. Trends Biochem Sci. 2017 mar; 42(3):171-9)

3/ Page 9, legend of the Figure 2: Typo – “cross-oresnttion” should be “cross-presentation”

Author Response

Response to Reviewers’ Comments

Reviewer 2:

We wish to thank the reviewer for his/her positive evaluation and his/her constructive suggestions. A point-by-point reply to the reviewer’s comments is as follows:

Page 5, para 6: Here, the authors should also discuss the fact that antigen export is also influenced by the size of the phagocytosed antigen ((see for example this publication: Mant et al. Immunology. 2012 jun; 136(2):163-75).

Response: The fact that antigen export is also influenced by the size of the phagocytosed antigen has been discussed, as suggested (please see page 5–6, line 248–250).

Page 6, para 6: In this section, the authors should be more moderated concerning the contribution of Sec61 in antigen cross-presentation. The precise involvement of Sec61 remains unclear and fraught with technical issues (for discussion, see this publication: Romisch K. Trends Biochem Sci. 2017 mar; 42(3):171-9).

Response: We pointed out that the precise involvement of Sec61 in CP remains unclear and that further experimentation aimed at clarifying the role of the SEC61 complex for CP is needed (please see paragraph 6.2.3).

Page 9, legend of the Figure 2: Typo – “cross-oresnttion” should be “cross-presentation”.

Response: We thank the reviewer for the careful revision. The indicated typo has been corrected.

This manuscript is a resubmission of an earlier submission. The following is a list of the peer review reports and author responses from that submission.

Round 1

Reviewer 1 Report

The authors of the above paper attempt to give a comprehensive review of the up-to-date knowledge on the ability of dendritic cells to cross-present exogenous antigens to MHC class I molecules. Mainly focusing on molecular mechanisms of the process and distinct cellular compartments where self or non-self antigens are loaded onto MHC I.

However, the manuscript is poorly written, the English is rigged with multiple linguistic and stylistic errors making it difficult to follow and understand what authors meant. There is also flawed logical progression of arguments. The order of introducing concepts and then explaining them in depth is confused. For example: The introduction (section 1) didn’t really set the scene for the review – should give a broad summary and tell the reader why it is important, and what has been advanced since the last review in the subject area. Then section 3 and 4 would fit better as an introduction. Sections should be re-ordered.

As a systematic review a reader has to be guided through a complex story with a strong focus on one clear aspect at a time. However, here there are multiple repetitions of the same concepts in sequential slightly different sentences. It gives impression of jumping from aspect to aspect and then repeating itself to regain the context. There are aspects that are introduced without a context and then – “so what?” does not follow. Examples: lines 44, 99, 106, 170-172, 177, 219, 222, 282-285, 291, etc.

The only two sections that were relatively easier to follow were section 4 and 6.

There are wrongly used plurals and “upon” that occlude the understanding of the message. There are also some unfortunate uses of words. Examples: lines 33, 38, 41, 47, 50, 58, 59, 64, 68, 81, 90, 98, 107, 174, 183, 288, 341, 344 etc.

The story is jumping between human and mouse DCs, mo-DCs vs steady-state DCs without clarifying which and where is talked about.

There are instances where recent studies are mentioned and cited 2005-2011 which are still 7 years ago… In the entire manuscript there is a reference to only 1 paper that was published last year – defies the purpose of a review to collate up-to-date knowledge.

This review in the current form does not add much to the DC cross-presentation scene, other than reintroducing a good work of the authors that was published in 2016 and 2017 – even the Figure 2 is only slightly updated. In my opinion it does not further the knowledge and deems unnecessary the purpose of a review. Particularly in a context of recently published reviews in this area:

Nataschja, Ho et al. “Adjuvants Enhancing Cross-Presentation by Dendritic Cells: The Key to More Effective Vaccines?” Frontiers in immunology vol. 9 2874. (2018).

Cruz, Freidrich M et al. “The Biology and Underlying Mechanisms of Cross-Presentation of Exogenous Antigens on MHC-I Molecules.” Annual review of immunology vol. 35 (2017): 149-176.

Gutiérrez-Martínez, Enric et al. “Cross-Presentation of Cell-Associated Antigens by MHC Class I in Dendritic Cell Subsets.” Frontiers in immunology vol. 6 363. (2015).

I would strongly suggest to re-write the entire manuscripts, introducing systematic and clear progression of logical arguments and highlighting the impact and how the recent studies, including last year, have advanced the field.

Reviewer 2 Report

In this manuscript, Imai et al review recent advances on antigen cross-presentation, focusing on the intracellular transport routes for exogenous antigens and the distinctive subcellular compartments involved in the endosplasmic reticulum associated degradation. The paper is clearly written, interesting and timely.

I have a few specific points to address:

1/ Page 5, para 6: Here, the authors should also discuss the fact that antigen export is also influenced by the size of the phagocytosed antigen ((see for example this publication: Mant et al. Immunology. 2012 jun; 136(2):163-75).

2/ Page 6, para 6: In this section, the authors should be more moderated concerning the contribution of Sec61 in antigen cross-presentation. The precise involvement of Sec61 remains unclear and fraught with technical issues (for discussion, see this publication: Romisch K. Trends Biochem Sci. 2017 mar; 42(3):171-9)

3/ Page 9, legend of the Figure 2: Typo – “cross-oresnttion” should be “cross-presentation”